# The cost-effectiveness of a two-step blood pressure screening programme in a dental health-care setting

Helen Andersson[1,2]*, Mikael Svensson[2], Håkan Bergh[2,3]

**1** Hallands Hospital Varberg, Varberg, Sweden, **2** School of Public Health and Community Medicine, Institute of Medicine, University of Gothenburg, Gothenburg, Sweden, **3** Dept. of Research & Development Unit, Hospital Varberg, Region Halland, Varberg, Sweden

* Helen.A.Andersson@regionhalland.se

**Data Availability Statement:** All relevant data are within the manuscript and its Supporting Information files.

**Funding:** This work was supported with grants from the County Council of Halland, Sweden, and

## Abstract

### Background

Hypertension is one of the largest contributors to the disease burden and a major economic challenge for health-care systems. Early detection of persons with high blood pressure can be achieved through screening and has the potential to reduce morbidity and mortality. We evaluate the cost-effectiveness of an opportunistic hypertension screening programme in a dental-care facility for individuals aged 40–75 in comparison to care as usual (the no-screening baseline scenario).

### Methods

A cost-effectiveness analysis (CEA) was carried out from the payer and societal perspectives, and the short-term (from screening until diagnosis has been established) cost per identified case of hypertension and long-term (20 years) cost per quality-adjusted life year (QALY) were reported. Data on the short-term cost were based on a real-world screening programme in which 2025 healthy individuals were screened for hypertension. Data on the long-term cost were based on the short-term outcomes combined with modelling in a Markov cohort model. Deterministic and probabilistic sensitivity analyses were carried out to assess uncertainty.

### Results

The short-term analysis showed an additional cost of 4,800 SEK (€470) per identified case of hypertension from the payer perspective and from the societal perspective 12,800 SEK (€1,240). The long-term analysis showed a payer cost per QALY of 2.2 million SEK (€210,000) and from the societal perspective 2.8 million SEK per QALY (€270,000).

### Conclusion

The long-term model results showed that the screening model is unlikely to be cost-effective in a country with a well-developed health-care system and a relatively low prevalence of hypertension.

Sparbanksstiftelsen, Varberg. The funders had no role in study design, data collection and analysis, decision to publish, or preparation of the manuscript.

**Competing interests:** The authors have declared that no competing interests exist.

## Introduction

Hypertension or high blood pressure (BP) is an important worldwide public health problem and the most important risk factor for the total disease burden worldwide [1], with its sequelae including stroke and myocardial infarction [2]. It is estimated that 10% of health-care spending is directly related to hypertension and its complications [3]. The overall prevalence of hypertension in adults is approximately 25–45% in Europe [4]. In Sweden, high BP affects an estimated 1.8 million people, representing 27% of the adult population (the prevalence increases with age from 12% in young adults to 56% in the elderly) [5]. Since there are effective treatments that reduce both high BP and an individual's risk of developing sequelae [6], it is important to identify those individuals who have high BP as early as possible. Early detection of individuals with high BP may be done through BP screening among "healthy" individuals.

One type of screening is opportunistic screening, whereby a patient utilizes a health-care facility for another reason and, in addition to the regular treatment related to the visit, receives BP screening. Since a majority of the population (80% in Sweden) regularly seeks dental-care services in the form of annual check-ups [7], dental-care service providers can be a possible provider of screening for hypertension, as shown in several studies [8–10].

There is a knowledge gap on the optimal population screening programme for detecting hypertension [2], and there is a great need to evaluate the long-term cost-effectiveness of such programmes. One such initiative, an opportunistic two-step screening of hypertension, was tested during the dental-care visits of a general population, resulting in a positive predicted value of 0.76 and a reduction of the false positive values by 86% via a second step of home BP measurement [10]. The cost-effectiveness of an opportunistic screening programme for high blood pressure in a general population has not previously been assessed. We have used the results from the opportunistic two step screening of hypertension [10] to conduct a follow-up cost effectiveness analysis to address this question. Thus, the aim of this study was to evaluate the cost-effectiveness of the aforementioned opportunistic two-step hypertension screening programme.

## Methods

### The intervention

A two-step BP screening was conducted at four different dental clinics in a region of southern Sweden. The intervention was a single-arm screening programme implemented in an unscreened population, and the no-screening comparator group in this evaluation is assumed to be characterized by the status quo in which blood pressure tests are carried out when individuals visit health-care facilities. In the screening, BP was measured after five minutes of rest by a dental nurse twice in both arms (first step), and those with a mean BP value $\geq 140$ and/or $\geq 90$ mmHg were asked to use a home blood pressure device (Omron M6 Comfort) for one week (twice in the morning and in the evening) (second step). If the home BP resulted in a mean value $\geq 135$ and/or $\geq 85$, the individuals were referred to a primary health-care centre (PHC) for further assessment and diagnosis.

Both written and oral consent was obtained and the study is approved by the ethical review board in Lund, No. 2013/553 and 2015/446.

### Cost-effectiveness analysis

The analysis evaluates the two-step screening programme compared to the no-screening baseline in terms of short-term (from screening until diagnosis has been established, approximately 1–3 months) and long-term (20 years) outcomes. The short-term analysis uses an intermediate

outcome measure, identified hypertension patients, and the long-term analysis uses quality-adjusted life years (QALYs) as the outcome metric. The result is presented in terms of the incremental cost-effectiveness ratio (ICER), which is the difference in costs divided by the difference in health outcomes with the screening programme compared to no-screening baseline scenario: $(Cost_{SCREENING} - Cost_{NO\ SCREENING})/(Outcome_{SCREENING} - Outcome_{NO\ SCREENING})$.

Sub-group analyses of the screening programme based on sex are carried out considering the sex differences in the incidence of hypertension-related conditions, especially acute myocardial infarction (AMI). Moreover, cost-effectiveness is evaluated from a societal as well as a payer perspective. The difference between the two perspectives is that the societal perspective also includes the costs of the programme for the included individuals (primarily time-use and travel-related costs). All costs are expressed in 2019 prices (consumer price index adjusted) [11] in Swedish kronor (SEK), and the main results are also presented in euro (EUR) assuming an exchange rate of 1 EUR = 10.3 SEK (July 2020) [12]. The economic evaluation model was built and analysed in Microsoft Excel [13] and Stata v.16 [14].

## Short-term analysis

The short-term analysis includes the time frame up until persons are potentially diagnosed with hypertension and thus estimates the ICER in terms of the cost of identifying one patient with hypertension through the screening programme.

The model relies on the data from the primary screening study [10]. The formal dental and health-care cost data include the blood pressure test costs in dental and primary health-care facilities, ECG costs, and laboratory and diagnostic costs (Table 1). Non-health-care costs include patient time costs and travel costs [15]. Patient time costs refer to the time spent on the blood pressure tests in the dental-care setting and, for patients referred to the PHC, also the time spent in this latter setting. We assume that the visits did not displace working hours for the patients and thus value each hour of patient time based on average net wages [16]. Travel costs (to the PHC) are based on the average distance (3 km) and a cost of 1.85 SEK per km.

The number of newly discovered cases of hypertension in the screening group (170 individuals) is compared with a corresponding number in a hypothetical comparator arm (46 individuals). The parameter value in the comparator arm is based on an assumption that 61 (expected incidence 3%) [17, 18] of the 2025 individuals would have been identified as having high blood pressure during a visit to a primary care centre for some reason (on the patient's own initiative, at a doctor's suggestion, or for other reasons).

Since the diagnosis of hypertension was based on repeated blood pressure measurements both at home and in clinic (screening arm), it was assumed that no one in the screening arm was false positive.

Among those diagnosed with high blood pressure based on blood pressure screening in a clinical setting (comparing arm), we estimate that approximately 15 individuals (25%) present false positives (that is, they display white coat hypertension [WCHT]), and the remaining 46 (75%) are expected to be true positives [19].

## Long-term analysis: Markov-cohort model

For long-term costs and health outcomes, we developed a Markov-cohort model with the structure shown in Fig 1. At the time of the introduction of the screening programme, the entire cohort is in the "Healthy" state, which is also the status quo of the comparator case without the screening programme. Consequently, there are annual (one-year-cycle) risks of an AMI or stroke incident based on the risk equations from the Framingham studies adjusted for age, sex, lipid levels, and diastolic blood pressure [20]. There is also an annual age- and sex-

**Table 1. Input data on costs and health outcomes.**

| | Value | Uncertainty range | Distribution | Reference |
|---|---|---|---|---|
| **General parameters** | | | | |
| Cohort size | 2025 | Fixed | | [10] Andersson et al. (2017) |
| Cohort men | 930 | Fixed | | [10] Andersson et al. (2017) |
| Cohort women | 1095 | Fixed | | [10] Andersson et al. (2017) |
| Discount rate | 3% | 0–5% | Uniform | |
| Hypertension outcomes | According to original study | | Normal | [10] Andersson et al. (2017) |
| **Transition probabilities** | | | | |
| Stroke & AMI risks | According to Framingham risk equations, | | Beta | [20] Andersson et al. (1991) |
| Non-stroke/AMI mortality risks | According to Swedish population life tables | | Beta | [26] SCB 2017 |
| 365-day stroke mortality | Men: 0.102 | ±20% | Beta | [26] SCB 2017 |
| | Women: 0.144 | | | |
| 365-day AMI mortality | Men: 0.144 | ±20% | Beta | [26] SCB 2017 |
| | Women:0.173 | | | |
| Added mortality risk (+365 days) after stroke | Men: 0.074 | ±20% | Beta | [27] Eriksson et al. (2012) |
| | Women: 0.061 | | | |
| Added mortality risk (+365 days) after AMI | Men: 0.018 | ±20% | Beta | [28] Isaksson et al. (2011) |
| | Women: 0.017 | | | |
| **Formal care costs: short-term model** | | | | |
| Dental care BP test | 117 SEK | ±20% | Gamma | [15] Rapport RH (2017) |
| Health care BP test | 149 SEK | ±20% | Gamma | [15] Rapport RH (2017) |
| ECG | 75 SEK | ±20% | Gamma | [15] Rapport RH (2017) |
| Lab costs | 240 SEK | ±20% | Gamma | [15] Rapport RH (2017) |
| Diagnosis (identification) | 785 SEK | ±20% | Gamma | [15] Rapport RH (2017) |
| Screening program administration (30–40% of full time service) | 165,000 SEK | ±20% | Gamma | [15] Rapport RH (2017) |
| **Formal care costs: long-term model** | | | | |
| AMI costs first year | 112,000 SEK | ±20% | Gamma | [29] Lanitis et al. (2014) |
| Post-AMI costs | 2,670 SEK | ±20% | Gamma | [29] Lanitis et al. (2014) |
| Stroke costs first year | 112,000 SEK | ±20% | Gamma | [29] Lanitis et al. (2014) |
| Post-stroke costs (annual) | 85,000 SEK | ±20% | Gamma | [29] Lanitis et al. (2014) |
| Hypertension treatment costs (annual) | 2,150 SEK | ±20% | Gamma | [15] Rapport RH (2017) |
| **Informal costs: short-term model** | | | | |
| Patient time (per hour) | 160 SEK | ±20% | Gamma | |
| Patient travel cost | 6.5 SEK | ±20% | Gamma | |
| **Qaly-weight decrements** | | | | |
| Stroke | 0.50 | ±20% | Beta | [30] SBU 1994–2004 Tengs & Wallace (2001) |
| Post-stroke | 0.25 | ±20% | Beta | [30] SBU 1994–2004 Tengs & Wallace (2001) |
| AMI | 0.25 | ±20% | Beta | [31] SBU 1994–2004 Post et al. (2001) |
| Post-AMI | 0.05 | ±20% | Beta | [31] SBU 1994–2004 Post et al. (2001) |
| **Modeling assumptions** | | | | |
| Average SBP with hypertension | 147 | - | | |
| Average SBP with hypertension treatment | 140 | - | | |
| Average SBP without hypertension | 131 | - | | |
| Total cholesterol/HDL (Men) | 4.0 | - | | |
| Total cholesterol/HDL (Women) | 3.2 | - | | |

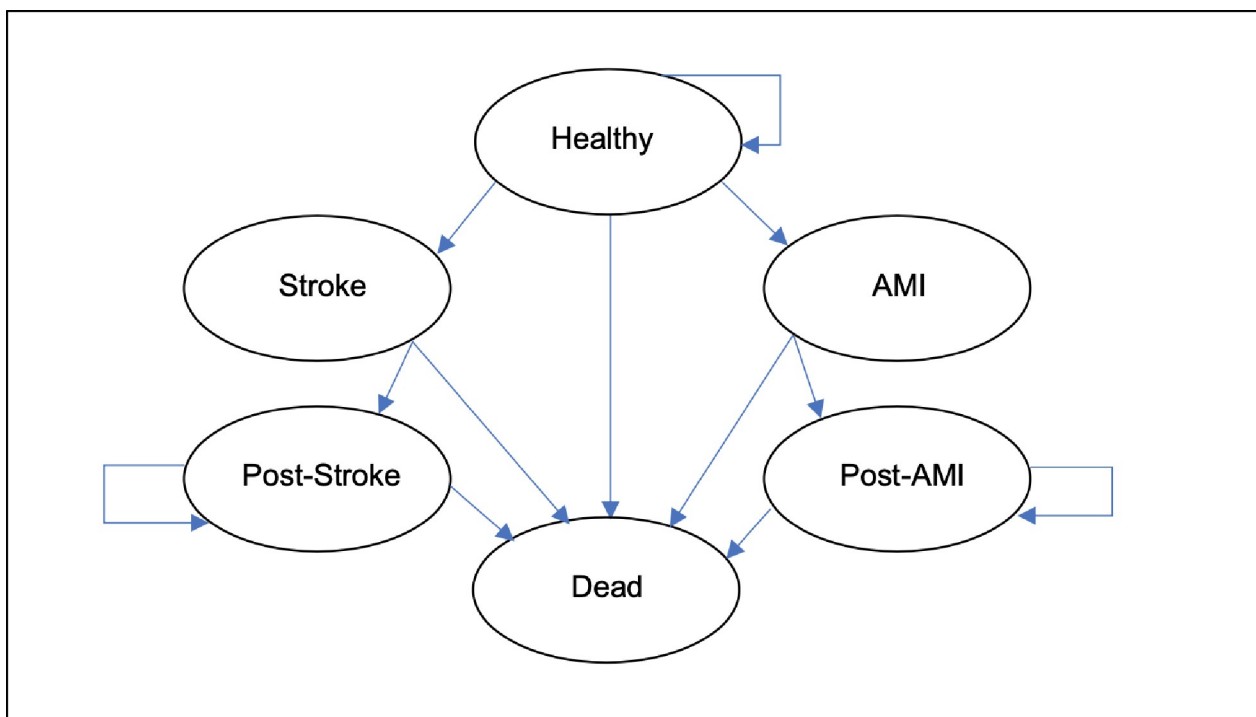

**Fig 1. Markov model structure.**

adjusted risk of mortality from other causes (not AMI or stroke) based on Swedish life-table data. From the AMI and stroke health states, there is either death as a direct consequence of the event or a transition to the post-AMI or post-stroke state. We make a simplifying assumption that there are no recurrent strokes or AMIs for the same person. The time perspective of the Markov model is 20 years, with annual discounting of costs and health outcomes of 3%, in line with the recommendations for cost-effectiveness analyses in Swedish health policy settings [21] (see Table 1 for input data on costs and transition probabilities). In this model based study the health outcomes are measured in terms of quality-adjusted life years (QALYs), which combine health-related quality of life (QALY weights) and life length [22]. QALY weights as used in the long-term Markov model are indexed such that 0 is interpreted as "equal to being dead" and 1 is interpreted as "the best possible health state". Table 1 lists the QALY-weight decrements, based on published evidence, associated with a stroke and AMI event.

## Assessing uncertainty

Parameter uncertainty analyses were carried out using (one-way) deterministic sensitivity analysis (DSA) and probabilistic sensitivity analysis (PSA) based on 5,000 Monte Carlo simulations. The results from the DSA are shown using a Tornado diagram where the ICER intervals are based on varying parameter input values for the time horizon of the Markov cohort model, underlying hypertension prevalence in the cohort, AMI and stroke costs, drug treatment costs, and QALY-weight decrements for AMI and stroke events.

The PSA assesses the uncertainty with jointly varying parameter values for hypertension prevalence, costs, transition probabilities, and QALY-weight decrements. The results from the PSA are shown using a cost-effectiveness plane (scatter-plot) and a cost-effectiveness

acceptability curve (CEAC), where the latter shows the probability that the screening programme (compared to the no-screening baseline) is cost-effective at different levels of the maximum willingness to pay per QALY ("threshold value").

The Swedish "threshold value" as stated by the National Board of Health and Welfare is that a cost per QALY is low if below 500,000 SEK, high if between 500,000 and 1 million SEK, and very high if above 1 million SEK [23]. All uncertainty ranges and distributions are listed in Table 1, with exception of the hypertension prevalence where the mean value of 170 identified persons with hypertension was associated with a standard error of 17.

## Results

Table 2 shows the short-term cost with and without the screening programme from a payer (health and dental care) as well as societal perspective. The increase in costs with the total screening programme is approximately 0.6 million SEK (€58,000) from a payer perspective and 1.6 million SEK (€154,000) from a societal perspective. The increase in costs in individual terms (total cost divided by the cohort size) with the screening programme is 295 SEK (€29) in a payer perspective and 785 SEK (€76) in a societal perspective.

As previously reported in the main publication on the screening programme [10], from the cohort of 2025 persons, mean age 52.8 (SD 8.7), the screening programme identified 170 (8%) persons as having true hypertension compared to an estimated 46 persons who would have been identified in the absence of the screening programme. The additional 124 persons correctly identified as having hypertension implies an incremental cost per identified hypertension case at approximately 4,800 SEK (payer perspective) and 12,800 SEK (societal perspective) (€470 and €1,240).

**Table 2. Short-term costs and health outcomes for the screening program vs the reference scenario.** Costs expressed in Swedish kronor, SEK (Euros in brackets for totals).

| Item | No Screening | Screening | Difference (Screening vs. No Screening) |
|---|---|---|---|
| **Health and dental care costs***  | | | |
| Fixed screening program cost | | 165,343 | 165,343 |
| BP test (dental care) | | 236,925 | 236,925 |
| BP test (primary care) | 47,978 | 120,988 | 73,010 |
| ECG | 4,575 | 12,750 | 8,175 |
| Lab tests | 14,640 | 40,800 | 26,160 |
| Setting diagnosis | 47,824 | 133,280 | 85,456 |
| *A. Total health and dental care costs* | 115,017 (€11,167) | 710,086 (€68,940) | 595,069 (€57,774) |
| **Non-health and dental care costs***  | | | |
| Time use (dental care) | | 162,000 | 151,875 |
| Time use (primary care) | 29,280 | 194,880 | 155,250 |
| Time use (BP test at home) | | 656,320 | 615,300 |
| Travel costs | 1,586 | 9,685 | 7,476 |
| *B. Total non-health care costs* | 30,866 (€2,997) | 1,022,885 (€99,309) | 929,901 (€90,282) |
| *Total costs (societal perspective = A + B)* | 145,883 (€14,163) | 1,732,971 (€168,250) | 1,587,088 (€154,086) |
| **Health outcomes** | | | |
| True positive identified cases | 46 | 170 | 124 |
| **Cost-effectiveness results** | | | |
| Cost per identified case of hypertension (health care perspective) | 2,500 (€243) | 4,177 (€406) | 4,799 (€466) |
| Cost per identified case of hypertension (societal perspective) | 2,974 (€289) | 9,564 (€929) | 12,799 (€1,243) |

Note: 1 SEK = 1/10.3 EURO. *The unit cost for all cost items are listed in Table 1.

**Table 3. Long-term costs and health outcomes for a cohort of 2,025 individuals.** Costs expressed in Swedish kronor, SEK (Euros in brackets).

| | Incremental cost | Incremental QALYs | Incremental cost per QALY |
|---|---|---|---|
| **All** | | | |
| Societal perspective | 4.9 million (€475,000) | 1.77 | 2.8 million per QALY (€270,000) |
| Health/dental-care perspective | 3.9 million (€380,000) | 1.77 | 2.2 million per QALY (€210,000) |
| **Men** | | | |
| Societal perspective | 5.3 million (€515,000) | 3.18 | 1.7 million per QALY (€165,000) |
| Health/dental-care perspective | 4.4 million (€430,000) | 3.18 | 1.4 million per QALY (€135,000) |
| **Women** | | | |
| Societal perspective | 4.0 million (€390,000) | 0.66 | 6.1 million per QALY (€590,000) |
| Health/dental-care perspective | 3.0 million (€290,000) | 0.66 | 4.6 million per QALY (€445,000) |

Notes: Incremental cost and QALYs is the additional cost and QALYs with the screening program compared to without the screening program for a cohort of 2,025 individuals based on 3% annual discounting. The incremental cost per QALY is the additional cost for each gained QALY. Costs are rounded to the closest 100,000 SEK. QALY-differences between the programs were driven by differences in AMIs (1.5 less with the screening program) and Strokes (0.7 less with the screening program).

Table 3 shows the long-term costs and health outcomes in the full cohort as well as for men and women separately (assuming equal cohort size). The results show that the incremental cost with the screening programme is 3.9–4.9 million SEK (€380,000–€475,000). If we consider an all-male cohort, the incremental cost would be 4.4–5.3 million SEK (€430,000–€515,000), and in an all-female cohort, it would be 3–4 million SEK (€290,000–€390,000). The lower value in the range refers to the payer perspective, and the higher value refers to the societal perspective.

The QALY gain is estimated at 1.77 for the entire cohort but higher (3.18) if we assume an all-male cohort and lower (0.66) if we assume an all-female cohort. The better health outcomes for an all-male cohort are based on the higher hypertension prevalence as well as the higher (untreated) AMI risk among men.

The associated cost per gained QALY is approximately 2.2 million SEK (€210,000) in a payer perspective and 2.8 million SEK (€270,000) in a societal perspective. Considering an all-male cohort, the estimated results are 1.4 million SEK (€135,000) and in the societal perspective 1.7 million SEK (€165,000) per QALY. For an all-women cohort, the cost is estimated at 4.6 million SEK (€445,000) and 6.1 million SEK (€590,000) per QALY.

## Deterministic sensitivity analysis

Fig 2 shows the one-way deterministic sensitivity analysis when we vary the input parameter values for the model time horizon, QALY-weight decrements, AMI and stroke costs, drug treatment costs, and prevalence of hypertension. Substantial variations in the QALY-weight decrements, AMI and stroke costs, and drug treatment costs have only a modest impact on the estimated ICER. Instead, the analyses reveal that the major uncertainty comes from varying the prevalence of (undetected) hypertension in the screened population and the model time horizon. Assuming a higher prevalence in the screened population lowers the ICER (since this factor would improve the health gains from the screening programme and subsequent treatment), and a longer time horizon (30 years vs. 10 years) also improves the cost-effectiveness (lower ICER). The dashed vertical line represents a cost-effectiveness of 500,000 SEK per QALY (€48,500), which is often used as an informal threshold value in Swedish health policy, and as seen, the ICER never fall below that threshold value in any of the sensitivity analyses.

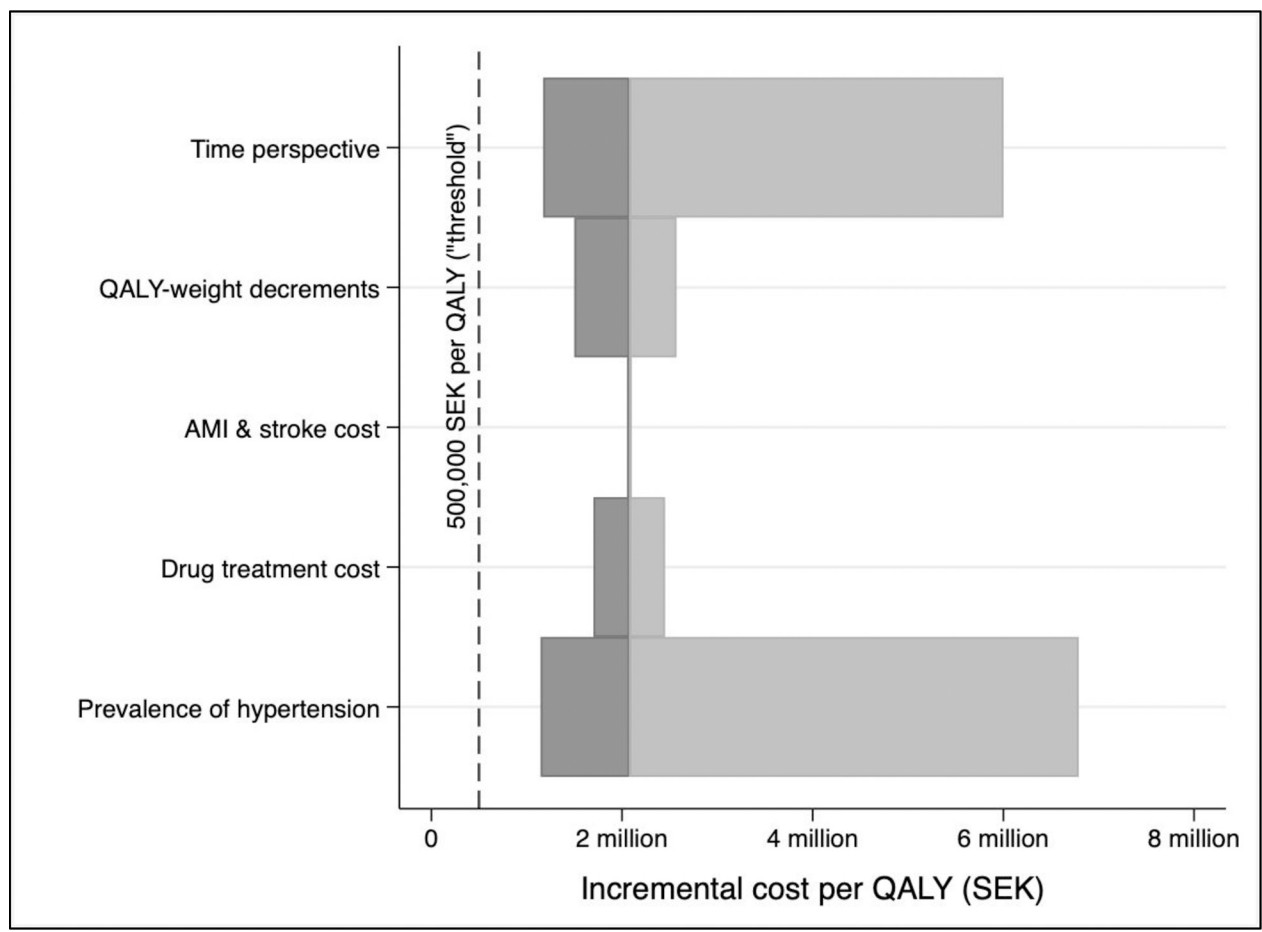

**Fig 2. One-way (deterministic) sensitivity analysis: Tornado diagram.** Notes: Input parameter values are varied one at a time (higher/lower) by 20% from the base-case assumptions and the ICER is then re-calculated in each case, except for the time-perspective, which varies 50% (higher/lower) compared to the base-case scenario. The bars show the lowest/highest ICER that is the result from each change in the input parameter values.

### Probabilistic sensitivity analysis

Fig 3 shows the result from the PSA in a scatter-plot (cost-effectiveness plane) with two different threshold-values included in the graph as well. Almost all ICERs are in the north-east quadrant of the cost-effectiveness plane, i.e. with higher costs and better health outcomes. None of the ICERs are below the 500,000 SEK per QALY threshold, and very few are below the 1 million SEK per QALY threshold. This can be seen more clearly in Fig 4, which shows the cost-effectiveness acceptability curve (CEAC) from the same data. The probability that the screening programme is cost-effective is approximately 0.02 at a willingness to pay per QALY of 500,000 SEK. At a willingness to pay per QALY of 1 million SEK, the likelihood that the screening programme is cost-effective is approximately 5%.

### Discussion

This is one of the few studies on the cost-effectiveness of screening for hypertension. The study model was built to capture the costs and outcomes of a programme for opportunistic screening of a general population in a "real-life" scenario, namely, an existing (dental-care) organization; such a setup has been recommended as a potential method of holding screening

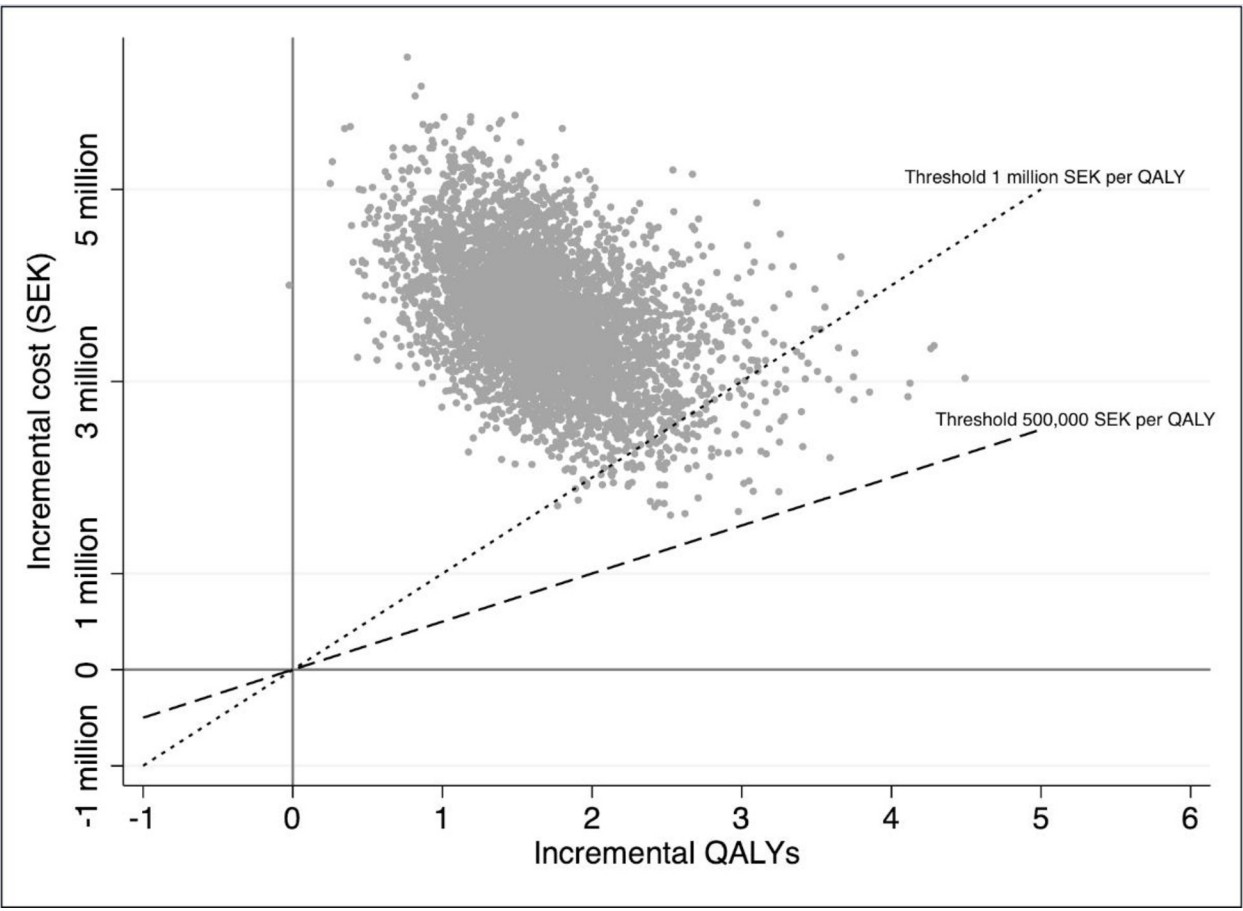

**Fig 3. Cost-effectiveness plane based on probabilistic sensitivity analysis.**

costs down. Blood pressure sampling was performed in the dental clinic, in the home environment, and in the PHC for at least 10 different days, which reduced the number of false positives by 85% [10]. The screening was performed on a previously unscreened population, which resulted in a large proportion (8%) of newly diagnosed cases being detected. Despite the above-mentioned good conditions, the model results for cost-effectiveness show a very high cost per gained QALY.

### Short-term analysis

The cost of the screening programme from the perspective of the dental- and health-care payer was 0.6 million SEK (€58,000) and with the addition of socio-economic costs rose to 1.6 million SEK (€154,000). The major cause of the difference in the two sets of costs is the inclusion of the patients' time cost of BP testing in the societal perspective.

The results of the short-term analysis show that the additional cost was approximately 4,800 SEK (€470) per newly discovered case in the form of dental- and health-care costs and approximately 12,800 SEK (€1,240) per newly discovered case with the inclusion of all societal costs. A similar study of a less effective opportunistic BP screening resulted in an NNS of 18, a PPV of 30%, and a direct cost of 5,300 SEK (€515) per newly discovered case [24].

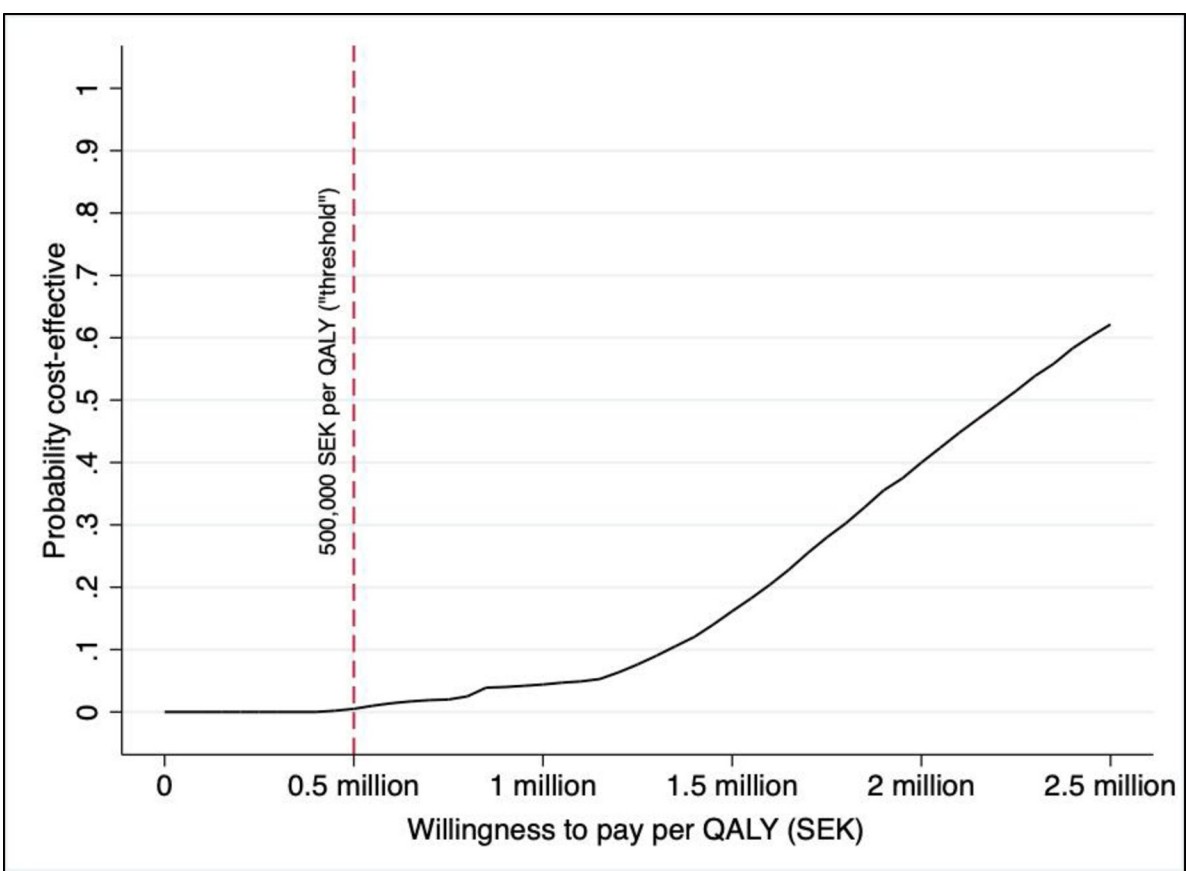

**Fig 4. Cost-effectiveness acceptability curve based on probabilistic sensitivity analysis.**

### Long-term analysis

The long-term consequences were analysed in a Markov cohort model, with the results show-ing a cost per QALY of approximately 2.2 million SEK (€210,000). When the patient's time cost was included to yield the societal perspective, the cost per QALY increased to over 2.8 mil-lion SEK (€270,000). This is substantially above the standard threshold values for the cost per QALY referenced in the Swedish health policy literature (SEK 500,000) [23]. In the sub-group analyses for men and women, the cost per QALY was lower for men (1.3–1.6 million SEK per QALY) than for women (4.3–5.7 million SEK per QALY). The lower cost per QALY in the cohort of men is primarily explained by the higher prevalence of AMI among men, especially in the relatively younger age groups (and thus a higher potential benefit of screening and drug treatment). However, even in an all-male cohort, the cost per QALY is above the standard threshold levels referenced in the Swedish health policy literature.

The sensitivity analyses show that the prevalence of hypertension and the time horizon have the greatest impact on the model's results. Sweden has a relatively low prevalence of hypertension (27%) and a well-developed health-care system, which means that many people with hypertension are already identified, which reduces the cost-effectiveness of adding screening. The time horizon in the baseline analysis was 20 years, and extending the horizon to 30 years improved the cost-effectiveness results somewhat (i.e., reduced the cost per QALY), though the cost remained above 1 million SEK per QALY.

The treatment of AMI and stroke has improved over time with improved survival, which also actually (relatively) reduces the value of screening and preventive treatment.

Further, most of the 170 newly identified persons had mild hypertension (grade 1), which can partly explain the high to very high cost per QALY from this screening programme. However, it should be noted that the health consequences considered from hypertension were limited to AMI and stroke. Should other consequences such as heart failure, renal failure, atrial fibrillation, cognitive impairment and dementia also be included, it is possible that the cost-effectiveness of the screening programme would be higher.

## Limitation

The transition probabilities (stroke and AMI risks) were based on risk models from the US Framingham study, and despite being widely used, they may have drawbacks in terms of validity for the given health context in this study [20]. As in all modelling-based studies, some simplifications had to be made that may have had some impact on the results. For example, an individual who has had an AMI can later suffer from stroke or vice versa, which our model did not allow for. And we have assumed that there is no difference between the two treatment alternatives in long-term identification of additional hypertension patients. An additional limitation with Markov cohort models is that average costs per, e.g., stroke and AMI, are assigned for each case and do not necessarily represent the costs in this particular cohort of patients.

The outcome in the comparator arm is based on the assumption that 61 individuals (expected incidence 3%) of the 2025 would have had high blood pressure during a routine consultation in a primary care centre and were diagnosed with hypertension (46 true positive and 15 false positive) [17, 18]. This assumption is based on results from previous screening studies that 50% of those with high blood pressure are newly diagnosed, and that white-coat hypertension can account for up to 25–40% of those with hypertension [2, 19]. We have chosen 25% for white-coat hypertension so as not to overestimate the result.

In the data set for our health economic analysis, there are no people with diabetes (as the condition was an exclusion criterion) and no information on cholesterol. The calculations refer to a population without diabetes. The mean values of serum cholesterol for men (4.0) and women (3.2) were used [25].

## Conclusions

Despite the success of a blood pressure screening programme in identifying a substantial number of true positive hypertension patients in an existing dental-care facility, the cost per QALY was 2.2 million SEK (€210,000), which is considered a high cost. The results thus suggest that adding blood pressure screening in the dental-care setting is not cost-effective.

## Supporting information

**S1 Checklist.**
(PDF)

## Acknowledgments

The authors would like to thank the participants of the dental health and primary health clinics for their time and effort. They also send warm thoughts to Professor Björn Lindgren for his help with the design of the health economic analysis.

## Author Contributions

**Conceptualization:** Helen Andersson, Mikael Svensson, Håkan Bergh.

**Data curation:** Helen Andersson, Håkan Bergh.

**Formal analysis:** Helen Andersson, Mikael Svensson, Håkan Bergh.

**Funding acquisition:** Helen Andersson, Håkan Bergh.

**Investigation:** Helen Andersson, Mikael Svensson, Håkan Bergh.

**Methodology:** Helen Andersson, Mikael Svensson, Håkan Bergh.

**Project administration:** Helen Andersson.

**Resources:** Helen Andersson, Håkan Bergh.

**Supervision:** Mikael Svensson, Håkan Bergh.

**Validation:** Mikael Svensson, Håkan Bergh.

**Visualization:** Helen Andersson, Mikael Svensson, Håkan Bergh.

**Writing – original draft:** Helen Andersson, Mikael Svensson, Håkan Bergh.

**Writing – review & editing:** Helen Andersson, Mikael Svensson, Håkan Bergh.

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
