## [Decision Letter · Decision Letter 0]

7 Jan 2021

PONE-D-20-30890

The cost-effectiveness of a two-step blood pressure screening programme in a dental health - care setting

PLOS ONE

Dear Dr. Helen Andersson,

Thank you for submitting your manuscript to PLOS ONE. After careful consideration, we feel that it has merit but does not fully meet PLOS ONE’s publication criteria as it currently stands. Therefore, we invite you to submit a revised version of the manuscript that addresses the points raised during the review process.

Can you also please complete a ‘Consolidated Health Economic Evaluation Reporting Standards (CHEERS)’ Statement indicating, where in the manuscript, each element of this statement has been addressed. Can you please also structure/format the manuscript according to this statement.

We look forward to receiving your revised manuscript.

Kind regards,

Billingsley Kaambwa

Academic Editor

PLOS ONE

Journal Requirements:

2. Please provide additional details regarding participant consent. In the Methods section, please ensure that you have specified (1) whether consent was informed and (2) what type you obtained (for instance, written or verbal). If your study included minors, state whether you obtained consent from parents or guardians. If the need for consent was waived by the ethics committee, please include this information.

5. Please include your tables as part of your main manuscript and remove the individual files. Please note that supplementary tables (should remain/ be uploaded) as separate "supporting information" files

Reviewers' comments:

Reviewer's Responses to Questions

**Comments to the Author**

1. Is the manuscript technically sound, and do the data support the conclusions?

Reviewer #1: Yes

Reviewer #2: Yes

Reviewer #3: Partly

2. Has the statistical analysis been performed appropriately and rigorously? 

Reviewer #1: Yes

Reviewer #2: Yes

Reviewer #3: I Don't Know

3. Have the authors made all data underlying the findings in their manuscript fully available?

Reviewer #1: Yes

Reviewer #2: Yes

Reviewer #3: Yes

4. Is the manuscript presented in an intelligible fashion and written in standard English?

Reviewer #1: Yes

Reviewer #2: Yes

Reviewer #3: Yes

5. Review Comments to the Author

Reviewer #1: Authors developed an economic model to assess the cost-effectiveness of an opportunistic hypertension screening programme compared to no screening of people aged 40-75 attending a dental-care facility. The authors found that under the current model structure and assumptions that opportunistic screening for hypertension in a dental care setting in a country with a relatively low prevalence is unlikely to be cost-effective.

In general, the manuscript was well written and structured. The abstract provides an accurate summary of the cost-effectiveness analysis. The conduct of the economic analysis conformed to best practices, but there some queries/suggestions.

• Given that the economic analysis considers the 40-74 years old population. Would it be helpful to include in the introduction the prevalence of hypertension in this population in Sweden?

• The intervention includes blood pressure readings taken by a dental health nurse, with some requiring home blood pressure measurements. Please can the authors elaborate on if participants were supplied with blood pressure monitors?

• Please can the authors provide the reference used to convert Swedish Kronor to Euros? It appears that some costs were obtained from the literature, please can the authors reference how these were inflated to current prices?

• Please can the authors state in the methods the software program that was used to undertake the economic analysis?

• The queries here relate to Table 1. First, the authors have included a cost of 165,000 SEK for vaccine program administration. Please can the authors elaborate on this cost included? Second, the authors state ‘formal care costs: short-term model.’ However, in my copy of the manuscript I have not seen this ‘short-term model’. Apologies if I have missed it. Third, it was not clear what was the average age of the population. Fourth, what utility value is used for people in the healthy state? Please can the authors expand on the abbreviations used in table 1?

• Based on the illustrative Markov model structure, it appears that people can only experience one event (e.g. cannot experience more than one stroke). Should this assumption be highlighted?

• Figure 1 presents the results of the one-way sensitivity analysis in the form of a tornado diagram. Please can the authors include the legend?

• I presume that with the two-step screening program, this will lead to early detection of people with hypertension compared to usual practice. Hence, it would be useful to show the impact of early detection, by reporting the number of events (e.g. stroke and AMI) that would be saved if this two-step program was implemented in practice.

• The queries here relate to the probabilistic sensitivity analysis (PSA). First, it was not clear what uncertainty range and distribution were used around the prevalence. Second, if space allows, please can the authors report the PSA results in the form of a scatterplot? Third, please consider re-phrasing the PSA results to, ‘The probability that the screening programme is cost-effective is approximately 0.02 at a willingness-to-pay threshold of 500,000 SEK per QALY.’

Reviewer #2: This is a well conducted study that is relevant in its field. Requirements for an economic evaluation according to the Drummond checklist were met.

The results could be presented in a less confusing way by indicating the ICER in both currencies for each perspective sequentially as opposed to putting them in brackets (page 1 line 32-35, page 6 line 200-204). Also indicate the currency year used for the Euro conversion.

Mention the time horizon in specific terms as opposed to simply indicating short-term and long term especially at first mention and in the abstract (page 1 line 24-25, page 3 line 95, 112)

Provide references for the model assumptions, especially those that are specific to the Swedish population (e.g page 4 line 130-133)

Minor comment: Regarding the Markov model, dead is an absorbing state so you do not expect a repeat arrow

Reviewer #3: This is a well written manuscript assessing the cost-effectiveness of a screening programme for hypertension in a dental health care setting compared to the "status quo". While I find the short term analysis easier to follow, further detail is required on the long term modelling approach so that I can assess the validity of the methods and the subsequent results. I provide some more specific comments below:

Short term analysis:

- Would be helpful to provide some more details on any potential false positives in the screening programme arm (or are there assumed to be none?)

- Data on the differential identificationis based on assumptions of the control arm. Please provide more details of the data used to inform these assumptions.

- It would be helpful to comment on the incremental analysis- it is not really screening programme or status quo, it is screening programme in addition to the status quo. How would this affect your analysis?

- What is the time horizon of the short term analysis? How many of the patients not identified in the control arm in this period would go on to be identified over the longer term by routine practice? What impact would this have on your results?

Long term analysis:

- Are there separate Markov models for those with identified hypertension, non identified hypertension and no hypertension or are you taking an average? The use of a markov model approach suggests a homogenous population entering the model at the beginning, as such if may not be a suitable approach if you do not model these 3 heterogeneous groups separately.

- Can unidentified patients in the model become identified?

6. PLOS authors have the option to publish the peer review history of their article (what does this mean?). If published, this will include your full peer review and any attached files.

Reviewer #1: No

Reviewer #2: No

Reviewer #3: No

---

## [Author Response · Author response to Decision Letter 0]

17 Feb 2021

Thank you for your most valuable comments on our manuscript. We found the comments useful and have made a revision of the paper. The proposed changes are marked in the manuscript and I have also specifically addressed each of the points made by the reviewers in the areas provided. I have uploaded my manuscript in a clean copy, the changed figures and tables. I have also uploaded a revised manuscripts, with markings that describe the changes I have made. Hopefully you are satisfied with our changes of the manuscript, tables and figures otherwise please contact us again so we can do further changes.

Best regards 

Helen Andersson

---

## [Decision Letter · Decision Letter 1]

20 Apr 2021

PONE-D-20-30890R1

The cost-effectiveness of a two-step blood pressure screening programme in a dental health - care setting

PLOS ONE

Dear Dr.****Andersson,

Thank you for submitting your manuscript to PLOS ONE. After careful consideration, we feel that it has merit but does not fully meet PLOS ONE’s publication criteria as it currently stands. Therefore, we invite you to submit a revised version of the manuscript that addresses the points raised during the review process.

We look forward to receiving your revised manuscript.

Kind regards,

Billingsley Kaambwa

Academic Editor

PLOS ONE

Journal Requirements:

Additional Editor Comments (if provided):

Page 3 line 18-21: This belongs to the results section

Page 3 line 25-29: Sentence is very long, Split sentence into two. Indicate the actual time for short-term in years or months

Page 3 line 30: The abbreviation is preceded by the word in full

Page 4 line 2-5: Indicate the source of costing data

Page 4 line 11-16: Indicate source of costing data and how non-health care costs were obtained

Page 4 line 38: write AMI in full at first time of mention

Page 5 line 9-11: Indicate how QALYs were obtained, was an instrument used, and frequency of QoL assessment

Page 5 line 29-30: Indicate the threshold value for Sweden

Results

Page 6 line 5: First time the link between this previous study and this present study is mentioned. This needs to be clearly spelt out in the introduction that this CEA is a follow-on from that previous study

Table 2: Include a column for unit costs

Reviewers' comments:

Reviewer's Responses to Questions

**Comments to the Author**

1. If the authors have adequately addressed your comments raised in a previous round of review and you feel that this manuscript is now acceptable for publication, you may indicate that here to bypass the “Comments to the Author” section, enter your conflict of interest statement in the “Confidential to Editor” section, and submit your "Accept" recommendation.

Reviewer #1: All comments have been addressed

Reviewer #2: (No Response)

Reviewer #3: All comments have been addressed

2. Is the manuscript technically sound, and do the data support the conclusions?

Reviewer #1: Yes

Reviewer #2: Yes

Reviewer #3: Yes

3. Has the statistical analysis been performed appropriately and rigorously? 

Reviewer #1: Yes

Reviewer #2: Yes

Reviewer #3: Yes

4. Have the authors made all data underlying the findings in their manuscript fully available?

Reviewer #1: Yes

Reviewer #2: Yes

Reviewer #3: Yes

5. Is the manuscript presented in an intelligible fashion and written in standard English?

Reviewer #1: Yes

Reviewer #2: Yes

Reviewer #3: Yes

6. Review Comments to the Author

Reviewer #1: The authors have responded to the concerns raised by the authors and have made the necessary changes to the manuscript.

Reviewer #3: The authors have fully responded to my comments and made appropriate ammendments to the paper. I have no further comments.

7. PLOS authors have the option to publish the peer review history of their article (what does this mean?). If published, this will include your full peer review and any attached files.

Reviewer #1: No

Reviewer #2: No

Reviewer #3: No

---

## [Author Response · Author response to Decision Letter 1]

7 May 2021

Thank you for your most valuable comments on our manuscript. We found the comments useful and have made a revision of the paper. The proposed changes are marked in the manuscript and I have also specifically addressed each of the points in the areas provided. I have uploaded my manuscript in a clean copy, the changed figures and tables. I have also uploaded a revised manuscripts, with markings that describe the changes I have made. Hopefully you are satisfied with our changes of the manuscript, tables and figures otherwise please contact us again so we can do further changes.

Best regards 

Helen Andersson

Response to Reviewers

-Page 3 line 18-21: This belongs to the results section. 

Comments: Thanks, we agree and have moved this sentence to the results section. Page 6 line 199-202

-Page 3 line 25-29: Sentence is very long, Split sentence into two. Indicate the actual time for short-term in years or months. 

Comments: Thanks, We have split the sentence, page 3 line 97-101 and we have clarified the actual time for short-term in months, page 3 line 99.

-Page 3 line 30: The abbreviation is preceded by the word in full. 

Comments: Thanks, We have made this change in line 103.

-Page 4 line 2-5: Indicate the source of costing data. 

Comments: We have clarified by indicate the source of costing data, ref 11, page 3 line 114.

-Page 4 line 11-16: Indicate source of costing data and how non-health care costs were obtained. 

Comments: We have clarified by indicate the source of costing data, ref 15, page 4 line 125.

-Page 4 line 38: write AMI in full at first time of mention. 

Comments: I have write AMI in full at fist time of mention in page 3 line 110.

-Page 5 line 9-11: Indicate how QALYs were obtained, was an instrument used, and frequency of QoL assessment

Comments: We have clarified this in page 5 line 159- 166. “In this model based study the health outcomes are measured in terms of quality-adjusted life years (QALYs), which combine health-related quality of life (QALY weights) and life length. QALY-weight decrements, is based on published evidence, associated with a stroke and AMI event”.

-Page 5 line 29-30: Indicate the threshold value for Sweden. 

Comments: We have clarified and indicate the threshold value for Sweden, page 5 line 181-183.

Results

-Page 6 line 5: First time the link between this previous study and this present study is mentioned. This needs to be clearly spelt out in the introduction that this CEA is a follow-on from that previous study. 

Comments: Thanks, we have clarified the link between the previous study and this present study in Introduction, page 2 line 75-76. “We have used the results from the opportunistic two step screening of hypertension to conduct this follow-up cost effectiveness analysis”...

-Table 2: Include a column for unit costs. 

Comments: The unit cost for all cost items are listed in table 1. We have clarified this with a note in table 2.

---

## [Editor Report · Decision Letter 2]

10 May 2021

The cost-effectiveness of a two-step blood pressure screening programme in a dental health - care setting

PONE-D-20-30890R2

Dear Dr. Helen Andersson,

We’re pleased to inform you that your manuscript has been judged scientifically suitable for publication and will be formally accepted for publication once it meets all outstanding technical requirements.

Kind regards,

Billingsley Kaambwa

Academic Editor

PLOS ONE
---

## [Editor Report · Acceptance letter]

17 May 2021

PONE-D-20-30890R2 

The cost-effectiveness of a two-step blood pressure screening programme in a dental health-care setting 

Dear Dr. Andersson:

I'm pleased to inform you that your manuscript has been deemed suitable for publication in PLOS ONE. Congratulations! Your manuscript is now with our production department. 

Kind regards, 

on behalf of

Dr. Billingsley Kaambwa 

Academic Editor

PLOS ONE